# A single dose of an adenovirus-vectored vaccine provides protection against SARS-CoV-2 challenge

Shipo Wu [1,3], Gongxun Zhong [2,3], Jun Zhang [1,3], Lei Shuai [2,3], Zhe Zhang [1], Zhiyuan Wen [2], Busen Wang [1], Zhenghao Zhao [1], Xiaohong Song [1], Yi Chen [1], Renqiang Liu [2], Ling Fu [1], Jinlong Zhang [1], Qiang Guo [1], Chong Wang [2], Yilong Yang [1], Ting Fang [1], Peng Lv [1], Jinliang Wang [2], Junjie Xu [1], Jianmin Li [1], Changming Yu [1], Lihua Hou [1✉], Zhigao Bu [2✉] & Wei Chen [1✉]

The unprecedented coronavirus disease 2019 (COVID-19) epidemic has created a worldwide public health emergency, and there is an urgent need to develop an effective vaccine to control this severe infectious disease. Here, we find that a single vaccination with a replication-defective human type 5 adenovirus encoding the SARS-CoV-2 spike protein (Ad5-nCoV) protect mice completely against mouse-adapted SARS-CoV-2 infection in the upper and lower respiratory tracts. Additionally, a single vaccination with Ad5-nCoV protects ferrets from wild-type SARS-CoV-2 infection in the upper respiratory tract. This study suggests that the mucosal vaccination may provide a desirable protective efficacy and this delivery mode is worth further investigation in human clinical trials.

[1] Beijing Institute of Biotechnology, Beijing, China. [2] State Key Laboratory of Veterinary Biotechnology, Harbin Veterinary Research Institute, Harbin, China. [3] These authors contributed equally: Shipo Wu, Gongxun Zhong, Jun Zhang, Lei Shuai. ✉email: houlihua@sina.com; buzhigao@caas.cn; cw0226@foxmail.com

The global coronavirus disease 2019 (COVID-19) pandemic has made the development of a vaccine a top global priority. There are more than 100 vaccine development projects in the WHO draft landscape of COVID-19 candidate vaccines, including viral vector-based vaccines, mRNA and DNA vaccines, subunit vaccines, nanoparticle-based vaccines and inactivated-whole virus vaccines[1]. Some COVID-19 vaccine candidates, including that based on inactivated vaccine, a chimpanzee adenovirus-vectored vaccine and a DNA vaccine, can significantly inhibit virus replication and protect non-human primates from SARS-CoV-2 pneumonia[2–4]. However, it may be easier for these vaccine candidates to protect against lower respiratory tract disease than against upper respiratory tract disease in challenge tests[2–4].

We chose a replication-defective human adenovirus type 5-based COVID-19 vaccine (Ad5-nCoV) that leverages our prior experience with the Ad5-vectored Ebola vaccine[5–7]. Ad5-nCoV encoding the full spike of SARS-CoV-2 has shown a favourable safety profile and immunogenicity profile with one vaccination in a phase 1 human clinical trial[8]. Considering that SARS-CoV-2 is a serious respiratory disease, the mucosal vaccination may lead to a better efficacy.

Here, we evaluate the protective efficacy of the mucosal vaccination route, in addition to the normal intramuscular vaccination route. The induction of antibodies, T-cell responses and protective efficacy of Ad5-nCoV against SARS-CoV-2 challenge following intramuscular and intranasal immunizations in wild-type BALB/c mice and ferrets have been examined.

## Results

**Construction and identification of Ad5-nCoV.** Antigen design was guided by prior knowledge[9], and the full spike (S) protein was selected as the immunogen based on the Wuhan-Hu-1 strain (YP_009724390). The S gene (14-1273 aa) was optimized for increased antigen expression in mammalian cells, and transgene expression was verified by western blot. The E1/E3 deleted replication-defective Ad5 encoding full-length S led by the tissue plasminogen activator (tPA) signal peptide (Ad5-nCoV) was confirmed as a vaccine candidate (Fig. 1a, b).

**Ad5-nCoV induce strong humoral and cellular immune responses.** The BALB/c mouse is an ideal animal model for SARS-CoV-2 infection both in the upper and lower respiratory tracts. The immune response induced by Ad5-nCoV was first evaluated in BALB/c mice. Six- to eight-week-old female BALB/c mice ($n = 10$ per group) received a single immunization of $5 \times 10^9$ virus particles (VP) (high dose), $5 \times 10^8$ VP (middle dose), or $5 \times 10^7$ VP (low dose) of Ad5-nCoV or $5 \times 10^9$ VP of the control vaccine (Ad5 vector) by the intramuscular (IM) or intranasal (IN) route at week 0. S-specific IgG, IgG1 and IgG2a antibody, anti-SARS-CoV-2-specific neutralizing antibody (NAb), IgA and cellular immune responses were detected in each group. ELISA IgG titres peaked at day 28 in the IM groups and then slightly decreased afterwards, while those in the IN groups remained at a steady peak from week 4 to week 8 (Fig. 1c, f). There were higher IgG titres in the high-dose IN groups than in the IM groups ($P < 0.0001$ at week 6, and $P = 0.0001$ at week 8) and no difference between the middle- or low-dose IM and IN groups ($P > 0.05$) at weeks 6 and 8 post immunization (Supplementary Fig. 1a). Both the IM and IN route could induced robust SARS-CoV-2 S-specific IgG1 and IgG2a response and the IM route achieved a significant higher ratio of IgG2a to IgG1 (Supplementary Fig. 2). NAbs were detected by a virus-specific microneutralization assay, and NAb titres reached peaks at week 6 or week 8 in the IN or IM group, respectively (Fig. 1d, g). NAb titres were significantly

higher in the high-dose IN groups than in the IM groups ($P < 0.0001$ at week 4 and week 6, and $P = 0.0021$ at week 8) from week 4 to week 8, while there was no difference in the middle-dose group at weeks 6 and 8 and no difference in the low-dose group at any time point (Supplementary Fig. 1b). The trend of SARS-CoV-2 pseudovirus NAb (PNAb) titres was similar to that of NAb titres (Fig. 1e, h; Supplementary Fig. 1c), and a relatively good correlation was shown among IgG titres, NAb titres and PNAb titres at week 6 and week 8 post immunization (Supplementary Fig. 3). S-specific IgG in the trachea-lung wash was detected in both the IM and IN groups at week 2 (Supplementary Fig. 4b) and week 10 (Supplementary Fig. 5a), but S-specific IgA was found only in the IN groups (Supplementary Figs. 4c, 5b). NAbs and PNAbs were also detected in the trachea-lung washes in both high-dose groups but not in either low-dose group (Supplementary Fig. 5c, d). Both the middle-dose IM and IN groups exhibited significant induction of IFNγ, TNFα and IL-2 responses in splenic CD8$^+$ T cells or CD4$^+$ T cells at week 2, with a higher level in the IM group than in the IN group (Fig. 1i, j). Dose-dependent cellular immune responses were found in the IM groups but not in the IN groups at week 10 (Supplementary Fig. 5e, f).

**Ad5-nCoV protect mice against SARS-CoV-2 infection.** Seven of ten vaccinated mice in every group were inoculated intranasally with a mouse-adapted SARS-CoV-2 virus (HRB26M) at a dose of $10^{3.6}$ plaque-forming units (PFU) per mouse at week 10 post immunization. Four and three out of seven mice in every dose group were sacrificed for viral load quantification in the lungs and turbinates at 3 and 5 days post inoculation (dpi), respectively. No virus was detected in the lungs or turbinates in all the IN vaccinated groups at 3 and 5 dpi by quantitative polymerase chain reaction (qPCR) and PFU assays, while all the IN control group animals were infected with a mean viral load of $1.2 \times 10^4$ PFU/g in the turbinates and $5.6 \times 10^5$ PFU/g in the lungs at 3 dpi (Fig. 2). No virus was detected in the lungs in all the IM-vaccinated groups at 3 and 5 dpi, while all the IM control group animals' lungs were infected with a mean viral load of $3.3 \times 10^6$ PFU/g at 3 dpi. Virus was detected in the turbinates of some mice of the IM group by PFU assays and qPCR, with a significant reduction in the high and middle-dose groups compared with the load of the IM control animals (Supplementary Fig. 6). These data demonstrated that a single low dose of Ad5-nCoV can completely protect both the upper respiratory tract and lungs of mice from infection and it seemed that the infection prevention in the upper respiratory tract is more difficult than in the lower respiratory tract from the different protection via different vaccination mode in mice.

**Ad5-nCoV protect ferrets from wild-type SARS-CoV-2 infection.** The ferret is a mammalian model in which SARS-CoV-2 can replicate efficiently in its upper respiratory tract with high virus loads, but not in the lungs[10,11]. We next evaluated the protective efficacy of the mucosal vaccination and IM vaccination of Ad5-nCoV in the upper respiratory tracts of ferrets, 18 ferrets were equally grouped into the IM vaccination group ($5 \times 10^{10}$ VP), the mucosal vaccination group (simultaneous oral delivery with $5 \times 10^{10}$ VP and IN delivery with $5 \times 10^{10}$ VP for one ferret) and the control group and challenged on week 4 after a single vaccination of Ad5-nCoV. All vaccinated ferrets produced S-specific serum IgG antibodies and NAbs at week 4 post vaccination, which were not detected in control animals, with no difference between the two vaccination groups (Fig. 3a, b). We also observed cellular immune responses in 5 out of 6 ferrets in the IM group and 3 out of 6 ferrets in the mucosal vaccination

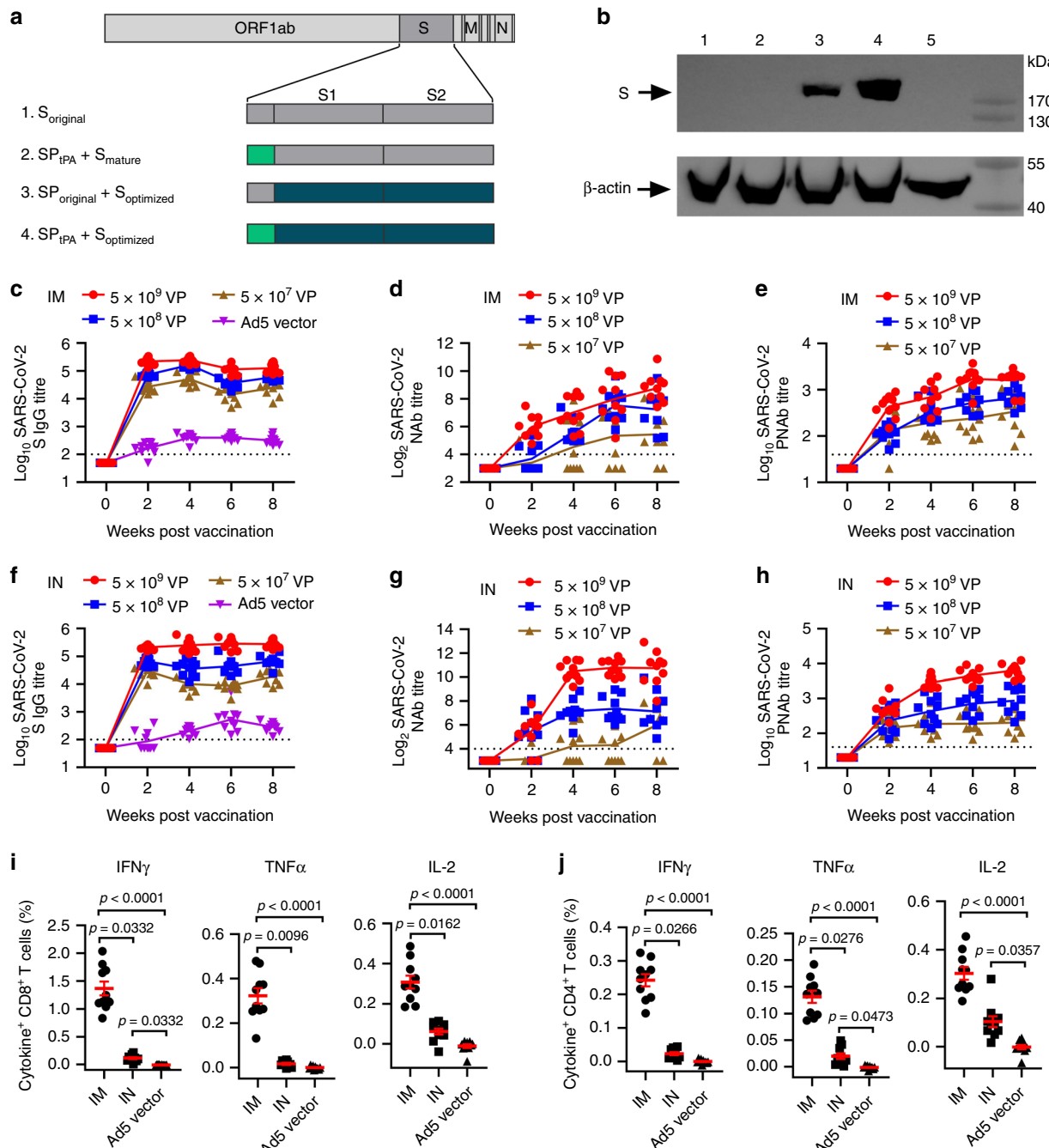

**Fig. 1 Adenovirus-based vaccine design and immunogenicity in mice. a** Schematic of SARS-CoV-2 S immunogens. **b** Western blot of transgene expression from (1) $S_{original}$, (2) $SP_{tPA}$ add $S_{mature}$, (3) $SP_{original}$ add $S_{optimized}$, (4) $SP_{tPA}$ add $S_{optimized}$, and (5) an empty plasmid transfected in HEK293 cells. BALB/c mice ($n = 10$ per group) received a single immunization with different doses of Ad5-nCoV or Ad5 vector by the IM or IN route. **c–h**, Humoral immune responses were assessed at weeks 0, 2, 4, 6 and 8 following vaccination by S-specific ELISA **c**, **f**, SARS-CoV-2 NAb titration ($MN_{50}$) **d**, **g** and SARS-CoV-2 PNAb titration **e**, **h** with $n = 10$ biologically independent animals per group. Data represent the individual titre of each animal and the connecting lines reflect the geometric means of the titres. **i**, **j**, Cellular immune responses were assessed at day 14 following vaccination in the $5 \times 10^8$ VP dose groups by intracellular cytokine staining assays with $n = 10$ biologically independent animals per group. Data are presented as mean ± s.e.m. Statistical significance was determined by Kruskal–Wallis ANOVA with Dunn's multiple comparisons tests. S = spike protein, SP = signal peptide, tPA = tissue plasminogen activator. Dotted line = the limit of detection. Source data are provided as a Source Data file.

group by IFNγ ELISpot assays at week 4 (Fig. 3c). These animals were challenged intranasally with $10^5$ PFU of SARS-CoV-2 at week 4 post vaccination, and nasal washes were collected every 2 days for viral load analysis of SARS-CoV-2 by qPCR and PFU assays. No virus was detected in the nose washes of mucosal vaccination group animals by the qPCR and PFU assays from 2 to

8 dpi, in contrast to all the infected control animals (Fig. 3d, e). Virus was detected in the nose washes of 3/6, 2/6, 0/6 and 0/6 of the IM-vaccinated animals by PFU assay at 2, 4, 6 and 8 dpi, respectively, and a significant reduction in viral load was found at 4, 6 and 8 dpi between the IM vaccination group and the control group animals (Fig. 3d, e).

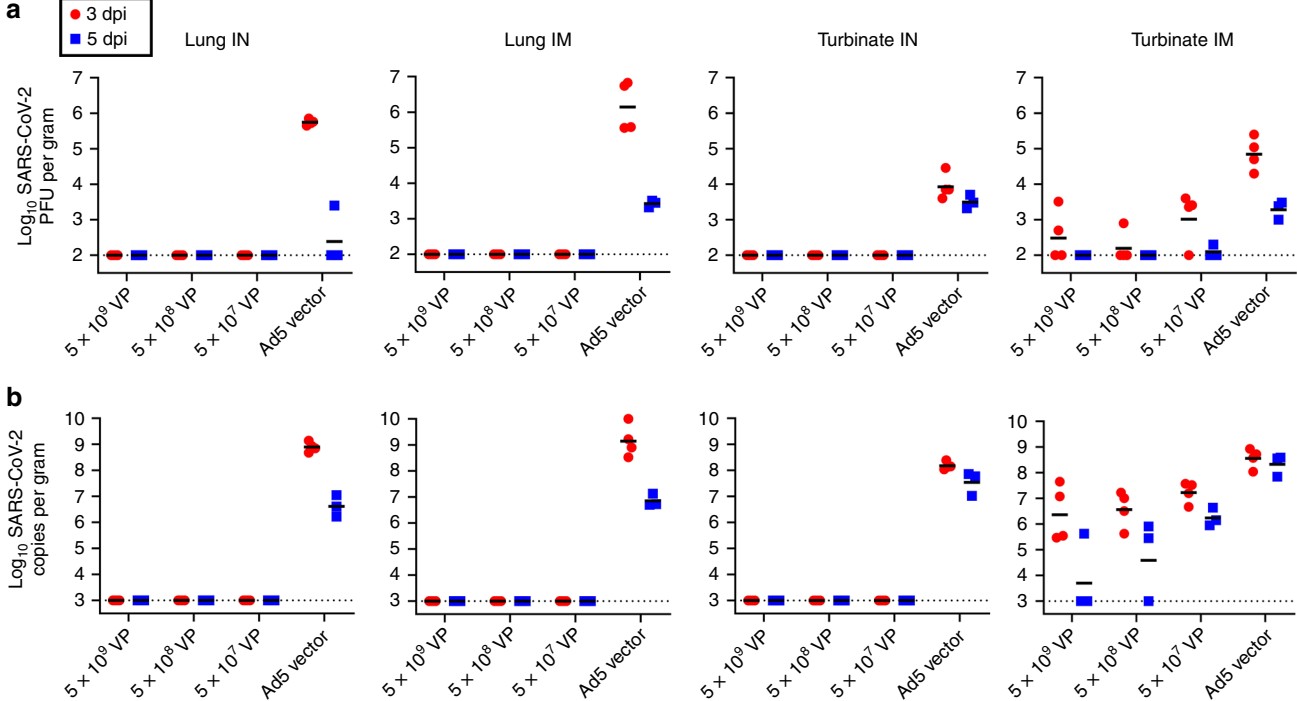

**Fig. 2 Protective efficacy of Ad5-nCoV in mice.** BALB/c mice ($n = 10$ biologically independent animals per group) received a single immunization with different doses of Ad5-nCoV or Ad5 vector by the IM or IN route, and seven of ten mice were challenged at week 10 with $10^{3.6}$ PFU of mouse-adapted SARS-CoV-2 virus HRB26M in a volume of 50 μL by the IN route. Tissue live virus numbers **a** and tissue viral loads **b** were detected at 3 dpi ($n = 4$ biologically independent animals per group) and 5 dpi ($n = 3$ biologically independent animals per group) in the lungs and turbinates, respectively. Black bars reflect geometric means; 3 dpi = red circles; 5 dpi = blue squares; dotted line = the limit of detection. Source data are provided as a Source Data file.

## Discussion

We have demonstrated for the first time, to our knowledge, that complete protection for the upper and lower respiratory tracts against SARS-CoV-2 infection can be achieved using a single mucosal inoculation of Ad5-nCoV in mice. A single intramuscular inoculation of Ad5-nCoV can protect the lungs of mice from SARS-CoV-2 infection and significantly reduce viral replication in the upper respiratory tract of mice and ferrets. Vaccine-elicited NAb titres are showed correlation with protective efficacy against SARS-CoV-2 infection in rhesus macaques[4]. The geometric mean of the serum NAb titres for the low-dose group of mice here is comparable to those of the subjects in the human clinical trial on day 28 post vaccination[8], and the serum NAb could be critical for the protection of the lower respiratory tracts.

The challenges administered to these animals with a high dose of virus via the intranasal route do not reflect realistic human exposure. Although the natural dose one might encounter is much lower than that of animals, the human is more permissive to SARS-CoV-2. Whether the IM vaccination or mucosal vaccination would protect the upper and lower respiratory tracts in humans under the natural exposure remains to be determined in phase III trials. The systemic humoral and cellular immune responses were strongly evoked when administered intramuscularly, which promoted the rapid defense against invading viruses. However the mucosal vaccination, which induces pathogen-specific mucosal immunity in addition to systemic immunity, can provide a first line of protection at the SARS-CoV-2 port of entry. The mucosal vaccine supply an important advantage that protects against virus replication in the upper respiratory tracts, interrupting person to person transmission. Although additional work is needed to fully unravel the protective mechanisms associated with the different vaccination delivery modes, we speculate that S-specific mucosal antibodies or vaccine-induced mucosal tissue-resident T cells play important roles in protection[12,13].

Pre-existing immunity to Ad5 is a common concern for the intramuscular delivery of Ad5 vector vaccines and the high pre-existing immunity did weaken the humoral and cellular immune response in some clinical trials[5,8,14]. However, the immune response by mucosal vaccination of Ad5-vectored vaccines may not be influenced by the high level of pre-existing immunity to Ad5[15–17]. One concern is that mucosal vaccination is complicated by the risk of inducing asthma attacks and some respiratory viruses are one of the most common causes of asthma exacerbations in both adults and children[18]. Live attenuated influenza vaccine given by nasal spray were safe and well-tolerated in children and adolescents with asthma in several clinical trials[19–21]. This suggests that it is very important for us to monitor the safety of mucosal delivery of Ad5-nCoV in people with asthma in the future clinical trials. In conclusion, this work gives us a hint that the different routes of vaccination should be considered in human clinical trials for the development of the SARS-CoV-2 vaccines.

## Methods

**Cells lines and viruses**. HEK293 cells (Human embryonic kidney, ATCC), Vero E6 cells (African green monkey kidney, ATCC) and ACE2-293T cells (ACE2-expressing cell line, constructed by hygromycin B screening) were maintained in Dulbecco's modified Eagle's medium (Thermo Scientific, USA) supplemented with 10% foetal bovine serum (Thermo Scientific, USA), penicillin (100 units/mL) and streptomycin (100 μg/mL) (complete medium) at 37 °C in 5% $CO_2$. SARS-CoV-2/human/CHN/Beijing_IME-BJ01/2020 (Genbank No. MT291831) and SARS-CoV-2/HRB25/human/2020/CHN (HRB25, GISAID access no. EPI_ISL_467430) were respectively isolated from patients and propagated in Vero E6 cells. Mouse-adapted SARS-CoV-2/HRB26/human/2020/CHN (HRB26M, GISAID access no. EPI_ISL_459910) was generated by passaging the human patient isolate SARS-CoV-2/HRB26/human/2020/CHN (HRB26, GISAID access no. EPI_ISL_459909) in 4–6-week-old (young) female mice for 14 passages and propagated in Vero E6

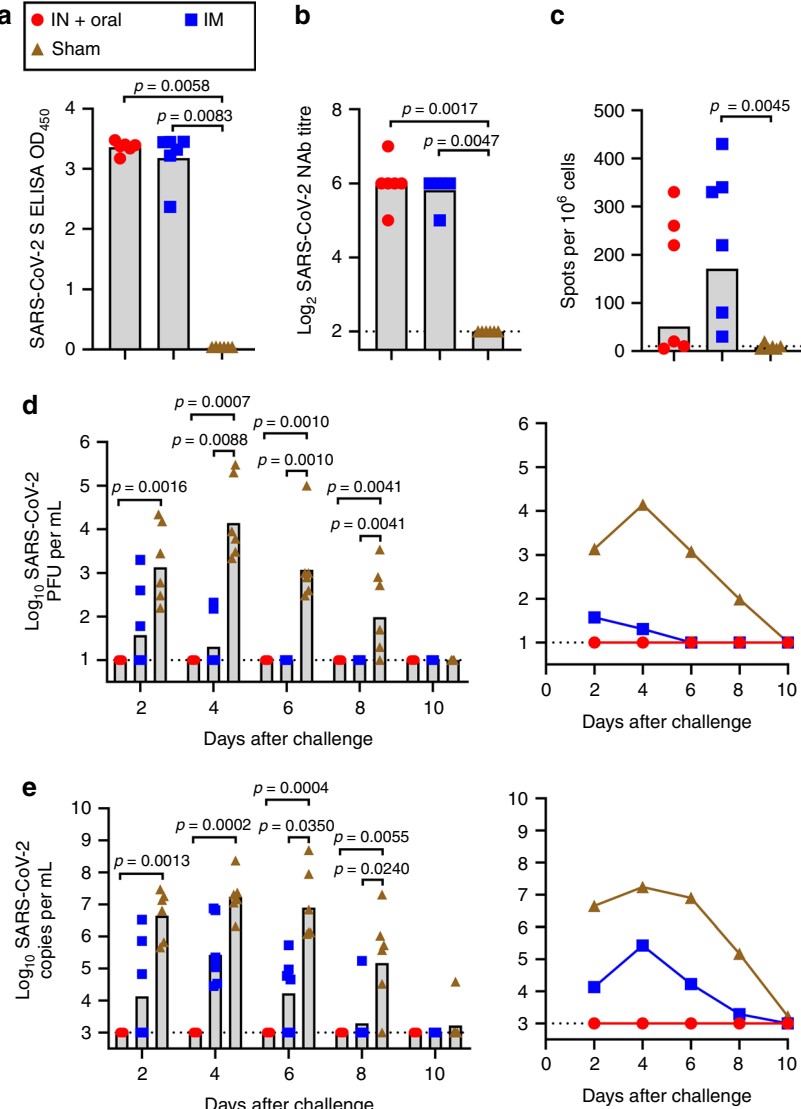

**Fig. 3 Immunogenicity and protective efficacy of Ad5-nCoV in ferrets.** Ferrets ($n = 6$ biologically independent animals per group) received a single immunization by the IM route or two simultaneous doses by the IN and oral routes with $5 \times 10^{10}$ VP per dose of Ad5-nCoV or received a sham vaccine. Humoral immune responses were assessed at week 4 following vaccination by S-specific ELISA (**a**) and NAb titration (PRNT$_{90}$) (**b**). **c** Cellular immune response was assessed by IFNγ ELISpot assays. Live virus numbers (**d**) and viral load (**e**) of SARS-CoV-2 in nasal washes obtained from ferrets after challenge. Bar at the geometric mean, and connecting lines reflect geometric means; statistical significance was determined by Kruskal–Wallis ANOVA with Dunn's multiple comparisons tests; IN and oral inoculation animals = red circles; IM injection animals = blue squares; control animals = brown triangles; dotted line = the limit of detection. Source data are provided as a Source Data file.

cells. Virus titres were determined by standard plaque assay on Vero E6 cells, and virus stocks were stored in aliquots at −80 °C until use.

**Construction of Ad5-nCoV expressing SARS-CoV-2 S.** The full spike protein gene of SARS-CoV-2 based on the Wuhan-Hu-1 strain (NC_045512.2) was codon optimized by UpGene software, and the signal peptide was substituted with tPA for increased expression in mammalian cells. The gene was synthesized with EcoRI and HindIII upstream and downstream of the open reading frame, respectively, and cloned into the shuttle plasmid of the AdMax adenovirus system (Microbix Biosystem, Canada) by enzyme digestion and ligation. After sequencing identification (sequencing primers can be found in Supplementary Table 1), the shuttle plasmid with the target gene was co-transfected into HEK293 cells with the backbone plasmid (pBHGloxΔE1, 3Cre) by TurboFect transfection reagent (Thermo Scientific, USA) according to the manufacturer's instructions. The transfected cells were passaged when they were overgrown and collected until Ad-related cytopathic effects were observed. The cells were lysed by three freeze-thaw cycles to release the recombinant viruses. The recombinant adenoviruses were confirmed by target gene sequencing, monocloned by agarose plaque selection, amplified by serial passage on HEK293 cells, and purified by ion exchange

chromatography and size exclusion. The number of total VP was measured by ultraviolet spectrophotometer analysis, with one OD$_{260}$ equal to ~$1.1 \times 10^{12}$ VP, and the infectious units were titrated on HEK293 cells using an AdenoX™ Rapid Titre Kit (Clontech, USA) following the manufacturer's instructions.

**Western blotting.** HEK293 cells in six-well cell culture clusters were transiently transfected with 2 μg of plasmids expressing different S protein constructs by TurboFect transfection reagent (Thermo Scientific, USA). The culture supernatant was discarded, and cells were rinsed with phosphate-buffered saline pH 7.4 (PBS) and lysed by 150 μL of RIPA Lysis and Extraction Buffer (Thermo Scientific, USA) for each well at 48 h post transfection. The cell lysate was centrifuged, and the supernatant was collected, mixed with NuPAGE LDS sample buffer (Thermo Scientific, USA) and run on a SurePAGE 4–20% Bis-Tris protein gel (GenScript, China) with 20 μL per lane. Protein was transferred to a nitrocellulose membrane by an eBlot L1 transfer system (GenScript, China). Membranes were blocked for 1 h at room temperature (RT) in Tris-HCl-buffered saline pH 7.6 with 0.1% Tween 20 (TBST) containing 5% skim milk and then incubated overnight at 4 °C with a 1:2000 dilution of a polyclonal rabbit anti-SARS-CoV spike antibody (Sino Biological, China). After four washes with TBST, the membranes were incubated for 1 h at RT with a

horseradish peroxidase (HRP)-conjugated goat anti-rabbit IgG antibody (Cell Signaling Technology, USA, 1:10,000 dilution). The membranes were developed with SuperSignal West Pico Chemiluminescent Substrate (Thermo Scientific, USA), and images were acquired with a Clinx ChemiScope imaging system (Clinx Science, China). As an internal parameter, β-actin was detected on the same membrane by an HRP-conjugated anti-β-actin antibody (Abcam, UK, 1:10,000 dilution).

**Animal experiments.** The experiments involving animals were approved by and carried out in accordance with the guidelines of the Institutional Experimental Animal Welfare and Ethics Committee. Specific pathogen-free female BALB/c mice aged 6–8 weeks were obtained from Beijing Vital River Laboratory Animal Technologies Co., Ltd (Beijing, China) and were housed and bred in the temperature-, humidity- and light cycle-controlled animal facility ($20 \pm 2$ °C; $50 \pm 10\%$; light, 7:00–19:00; dark, 19:00–7:00) of the Animal Center, Academy of Military Medical Sciences, Beijing. Three- to four-month-old female Angora ferrets purchased from Wuxi Cay Ferret Farm (Wuxi, China) were housed in the animal facility of Harbin Veterinary Research Institute (HVRI) of the Chinese Academy of Agricultural Sciences. Mouse and ferret experiments with infectious SARS-CoV-2 were performed at the biosafety level 4 and animal biosafety level 4 facilities in HVRI, which is approved for such use by the Ministry of Agriculture and Rural Affairs of China.

**Vaccination and challenge of mice.** BALB/c mice ($n = 10$ per group) were immunized intramuscularly or intranasally with $5 \times 10^9$ VP (high dose), $5 \times 10^8$ VP (middle dose) or $5 \times 10^7$ VP (low dose) of Ad5-nCoV at day 0 or $5 \times 10^9$ VP of Ad5 vector as a control. Sera were collected for S-specific ELISA, SARS-CoV-2 NAb titration ($MN_{50}$) and SARS-CoV-2 PNAb titration at different time points. Another three groups of mice immunized with a middle dose of Ad5-nCoV by the IM or IN route or with the control vaccine were euthanized at day 14 post immunization for splenic cellular immune response detection and trachea-lung wash antibody detection. At 10 weeks after vaccination, three of ten vaccinated mice in every group were euthanized for splenic T-cell response detection and trachea-lung wash antibody and NAb detection. The remaining mice (seven per group) were challenged intranasally with SARS-CoV-2 HRB26M strain at a dosage of $10^{3.6}$ PFU in a volume of 50 μL. Four and three out of seven mice in every dose group were sacrificed for viral load quantification in the lungs and turbinates at 3 and 5 dpi, respectively.

**Vaccination and challenge of ferrets.** Ferrets (6/group) were randomized by body weight, sex, and age and grouped into the IM vaccination group ($5 \times 10^{10}$ VP), the mucosal vaccination group (simultaneous oral delivery with $5 \times 10^{10}$ VP and IN delivery with $5 \times 10^{10}$ VP for one ferret) and the control group. Blood was collected at week 4 for analysis of antibody responses by ELISA, plaque reduction neutralization assays, and IFNγ ELISpot. The immunized animals were challenged intranasally with a dose of $10^5$ PFU of SARS-CoV-2 HRB25 strain at day 28, and the viral load of the nasal wash was detected by qPCR and PFU assay every 2 days post infection.

**Quantitative RT-PCR.** Viral loads were determined by quantitative real-time PCR. Viral RNA was extracted by using a QIAamp vRNA Minikit (Qiagen, Germany). Reverse transcription was performed by using HiScript II Q RT SuperMix for qPCR (Vazyme, China). qPCR was conducted by using an Applied Biosystems QuantStudio 5 Real-Time PCR System (Thermo Scientific, USA) with Premix Ex Taq for probe qPCR (TaRaKa, China). N gene-specific primers (forward, 5′-GGG GAA CTT CTC CTG CTA GAA T-3′; reverse, 5′-CAG ACA TTT TGC TCT CAA GCT G-3′) and probe (5′-FAM-TTG CTG CTG CTT GAC AGA TT-TAMRA-3′) were utilized according to the information provided by the National Institute for Viral Disease Control and Prevention, China (http://nmdc.cn/nCoV). The amount of vRNA for the target SARS-CoV-2 N gene was normalized to the standard curve from a plasmid (pBluescript II SK-N, 4,221 bp) containing the full-length cDNA of the SARS-CoV-2 N gene. The assay sensitivity was 1000 copies/mL.

**ELISA.** For SARS-CoV-2 S-specific IgG assays in mice, 96-well polystyrene high-binding microplates (Corning, USA) were coated with 2 μg/mL recombinant SARS-CoV-2 S protein purified from insect cells (Sino Biological, China) in carbonate-bicarbonate buffer pH 9.6, and the plates were incubated at 4 °C overnight. The plates were then blocked at 37 °C for 1 h with PBS pH 7.4 in 5% skim milk (blocking buffer) and washed with PBST. Serial dilutions of sera or trachea-lung washes of mice in dilution buffer were added to the plates and incubated at RT for 1 h. HRP-conjugated goat anti-mouse IgG, IgG1, IgG2a (Abcam, UK, 1:10,000 dilution) or HRP-conjugated goat anti-mouse IgA (Abcam, UK, 1:10,000 dilution) was added to the plates, and the plates were incubated at RT for 1 h and washed with PBST. The assay was developed for 10 min at RT with 100 μL of TMB substrate solution (Solarbio, China), stopped by the addition of 50 μL of stop solution (Solarbio, China) and then measured at 450 nm/630 nm (SPECTRA MAX 190, Molecular Device, USA). The endpoint titre was defined as the highest reciprocal serum dilution that yielded an absorbance ≥2.1-fold over negative control serum values.

A double antigen sandwich ELISA kit (ProtTech, China) was used for SARS-CoV-2 S-specific IgG assays for the ferrets. Briefly, 100 μL of serum was added to an antigen-coated microtitre plate, and the plate was incubated at RT for 30 min and washed with PBST. Then, the plate was incubated with HRP-conjugated antigen at 37 °C for 30 min and washed with PBST. The optical density (OD) was

measured at 450 nm after the addition of the substrate solution and the subsequent stop solution. Seropositivity was defined as an OD value ≥ 0.2.

**SARS-CoV-2 neutralization assay.** The neutralizing activity of sera from the mice was assessed using a microneutralization (MN) assay. Serial dilutions of heat-inactivated sera were incubated with 100 $TCID_{50}$ of SARS-CoV-2 IME-BJ01 strain at 37 °C for 2 h. Antibody-virus complexes were added to pre-plated Vero E6 cell monolayers in 96-well plates and incubated for 48~72 h. The cells were stained with 0.05% crystal violet for 30 min. The OD was measured at 570 nm/630 nm after the addition of the decolorization solution. Neutralization results were analyzed by Reed-Muench method to estimate the dilution of sera required for half-maximal neutralization of infection ($EC_{50}$ titre). The initial dilution of sera (1:16) was set as the limit of confidence of the assay. Seropositivity was defined as a titre ≥16.

The neutralizing activity of heat-inactivated sera from the ferrets was assessed using a plaque reduction neutralization test (PRNT) assay. Serial dilutions of sera were incubated with 50 PFU of the SARS-CoV-2 HRB25 strain at 37 °C for 21 h. Antibody-virus complexes were added to pre-plated Vero E6 cell monolayers in 24-well plates and incubated for 48 h with agarose overlay. NAb titres were calculated as the maximum serum dilution yielding a 50% reduction in the number of plaques relative to that for control serum prepared from uninfected animals. Seropositivity was defined as a titre ≥8.

**SARS-CoV-2 pseudovirus neutralization assay.** SARS-CoV-2 pseudovirus bearing the full-length spike protein of SARS-CoV-2 was produced in an Env-defective, luciferase-expressing HIV-1 backbone. A total of $7 \times 10^6$ 293 T cells were aliquoted into a 10-cm plate and co-transfected with 23 μg of pNL4-3.Luc-R$^-$E$^-$ and 1 μg of CMV/SARS-CoV-2-S by TurboFect transfection reagent (Thermo Scientific, USA). At 48 h later, the supernatants containing pseudovirus were collected, filtered, aliquoted and frozen at −80 °C. Serial dilutions of heat-inactivated sera were mixed with the titrated pseudovirus, incubated for 60 min at 37 °C and added to ACE2-293T cells in duplicate in 96-well microplate. Cells were lysed 48 h later and luciferase activity was measured. $EC_{50}$ neutralization titres were calculated for each individual mouse serum sample using Reed-Muench method.

**ELISpot.** SARS-CoV-2-specific cellular immune responses in ferrets were assessed by a Mabtech Ferret IFNγ ELISpot Kit (Mabtech, Sweden) following the manufacturer's instructions. In brief, $1 \times 10^5$ of peripheral blood mononuclear cells (PBMCs) of vaccinated ferrets were produced by density gradient sedimentation and stimulated with inactivated SARS-CoV-2 in a pre-coated ELISpot plate for 16 h in a 37 °C humidified incubator with 5% CO$_2$. The next day, the plate was washed five times with PBST, incubated for 1 h at RT with the biotin conjugated detection antibody, washed, incubated for 1 h at RT with streptavidin-HRP, washed, and developed with AEC substrate (BD Pharmingen, San Diego, CA). The plate was washed extensively in deionized water to stop colour development and dried in the dark, and the spots were counted in an AID ELISPOT reader (AID GmbH, Strassberg, Germany). The result was expressed as the number of SARS-CoV-2-specific spots per 1 million PBMCs.

**Intracellular cytokine staining.** Splenocytes of BALB/c mice were prepared by pushing the spleen through a 70-μm cell strainer, followed by red blood cell lysis and several washes. The cells were stimulated for 6 h at 37 °C with or without 1 μg/mL of overlapping 15-amino-acid peptides covering the S protein and with BD GolgiStop$^{TM}$ and BD GolgiPlug$^{TM}$ to block cytokine secretion. Following peptide pool stimulation, the splenocytes were washed and stained with a mixture of antibodies against lineage markers, including anti-CD3 PerCP-Cy5.5 (clone 17A2, 1:330 dilution), anti-CD4 Alexa Fluor 700 (clone RM4-5, 1:330 dilution), and anti-CD8 FITC (clone 5H10-1, 1:200 dilution), and the viability dye Near-IR to exclude dead cells from data analysis. After one wash with PBS, the cells were fixed and permeabilized with Cytofix/Cytoperm (BD Biosciences, USA), washed with Perm/Wash buffer (BD Biosciences, USA), and stained with anti-IFNγ PE (clone XMG1.2, 1:100 dilution), anti-TNF PE-Cy7 (clone MP6-XT22, 1:100 dilution) and anti-IL-2 Brilliant Violet 421 (clone JES6-5H4, 1:160 dilution). The cells were washed successively with Perm/Wash buffer and PBS and resuspended in PBS, and data were acquired on a FACS Canto$^{TM}$ (BD Biosciences, USA). At least 200,000 events were collected for each sample, and the data were analyzed by FACS Diva software. CD8$^+$ and CD4$^+$ T cells were gated from single cells (FSC-A vs FSC-H), lymphocytes (FSC-A vs SSC-A) and live CD3$^+$ T cells (CD3$^+$ vs Near-IR$^-$), successively, and the detection results were defined as the percentage of cytokine-positive cells among CD8$^+$ or CD4$^+$ T cells (Supplementary Fig. 7).

**Statistical analysis.** The analyses were performed with GraphPad Prism v.8.0.2. Two-tailed nonparametric Mann–Whitney's rank tests were conducted to compare differences between 2 experimental groups, Kruskal–Wallis ANOVA with Dunn's multiple comparisons tests were applied to compare >2 experimental groups. Antibody titre data were log transformed before analysis. Correlations were assessed by Spearman rank-correlation tests.

**Reporting summary**. Further information on research design is available in the Nature Research Reporting Summary linked to this article.

## Data availability

The authors declare that the data supporting the findings of this study are available within the paper and its Supplementary Information files, or are available from the corresponding author upon reasonable request. Source data are provided with this paper.

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

## Acknowledgements

We thank CanSinoBIO Inc. for providing Ad5-nCoV vaccine for this work. We thank Beijing Institute of Microbiology and Epidemiology for providing SARS-CoV-2 strain (SARS-CoV-2/human/CHN/Beijing_IME-BJ01/2020). We thank the support from the National Key Research and Development Program of China (2020YFC0841400, 2020YFC0846500), the National Major Science & Technology Major Project (2016ZX10004001, 2018ZX09201005).

## Author contributions

W.C., Z.B. and L.H. initiated and coordinated the project. S.W., Zhe Zhang, B.W., Zhenghao Zhao, Q.G., Jinlong Zhang and J.L. produced the vaccine candidate; Jun Zhang, L.F., X.S., Y.C., Y.Y., T.F., P.L., J.X. and C.Y. conducted the immunogenicity evaluation on mice. G.Z., R.L., L.S., Z.W., C.W. and J.W. conducted the immunogenicity evaluation on ferrets and the challenge studies on mice and ferrets. W.C., Z.B., S.W. and L.H. analyzed the data and wrote the manuscript.

## Competing interests

The authors declare no competing interests.
