## [Peer Review File · Nature Communications]

Reviewers' Comments:

Reviewer #1:

Remarks to the Author:

The paper uses ferrets infected with SARS-COV2 and mice infected with a mouse adapted SARS-COV2 to show rAD5 Spike SAR2-cov2 vaccine-mediated protection against challenge. This is a nice story and results look compelling. I have a number of minor suggestions.

- 1) The abstract should clarify that the mouse challenge used the mouse adapted virus, and that the ferret model (as I understand this) did not.
- 2) l24 Fig 2 uses a single vaccination. Fig 3 uses IM or IN + oral. Not clear where the i.m. i.n. combination data referred to in the abstract is to be found? If this is speculation would seem inappropriate (given the data set) especially in the abstract.
- 3) The notion of i.n. vaccination is complicated by the risk of inducing asthma attacks. Although the experience for influenza is perhaps promising for this route of administration (Paediatr Respir Rev. 2014 Dec;15(4):340-7) this risk might be discussed.
- 4) That preventing infection in the upper respiratory track is more difficult (l102) or easier in the lower respiratory track (l35) requires some clarity. More difficult in what sense? More antibodies needed? Virus replication not accessible to antibodies? T cells unable to engage MHC I? Higher vaccine doses needed to provide protection? Easier to access with vaccine? Higher replication capacity of virus? The concept is useful perhaps but "difficult" and "easy" are very vague and difficult to understand.
- 5) What is the rationale for in + oral? Oral vaccination needs some methodological explanation as quite an unusual route of entry? Any evidence that oral actually takes (works) in ferrets? What happens with i.n alone?
- 6) l40 perhaps rephrase "has shown a favourable safety profile"
- 7) l48 Very unclear what prior knowledge is referring to here, as no references provided. The rationale might be explained so that the reader can understand how this guiding worked.
- 8) Many have moved to chimp adenovirus because of pre-existing immunity to Ad5 – this needs at the very least to be discussed.
- 9) The MWU test assumes equal variance – a condition not met in several of the Figs eg Fig. 3 d and e where the red circles are all below detection (i.e. zero variance) and are compared with data sets that clearly have variance.
- 10) Would be helpful if the rAd5 doses used in humans were discussed to address whether the Ad5 doses used in the animal models might be viewed as appropriate?
- 11) l135 These were vaccine-induced tissue resident T cells – not just tissue resident T cells.
- 12) Did the lack of cellular immune responses l143/4 correlate with increased post change viral titers? Does this provide insights into the relative protective value of anti-viral T vs ab responses?

Andreas Suhrbier

Reviewer #2:

Remarks to the Author:

Comments for the Author

The Manuscript by Wu et al., describes the protective efficacy of Ad5-nCov vaccine against SARS-CoV-2 in mouse and ferret models. The current study demonstrated that upper respiratory tract (URT) administration of Ad5-nCov could achieve sterile protection against SARS-CoV-2 in both animal models. The manuscript also addresses a highly relevant point regarding the influence of vaccination route (i.e. IM vs IN route). The current study demonstrated that the vaccine administration to the URT

may be more effective in inducing immune responses at the site of SARS-CoV-2 infection. Overall, the work is well done and well presented. These data will provide important information to those currently developing SARS-CoV-2 vaccines.

Specific comments;

1: Authors should clarify the type of deletion that leads to the defect in replication in the results of the manuscript. Presumably this is E1/E3 deleted Admax?

2: The titers of neutralizing antibodies shown here for mice and ferrets are well above those published in their Lancet paper. This should be discussed as well as the potential impact of relative dose in humans compared to small animals.

3. It is probably worth mention in the discussion that a mucosal vaccine that protects against URT replication may have important advantages over parenterally administered vaccines with regard to interrupting person to person transmission.

4. Line 131-"Whether a lower challenge dose would result....."-this may be true but the opposite result is equally likely. Although the natural dose one might encounter is lower, the human is more permissive to SARS-CoV-2. It is likely worth noting both possibilities and that we need to wait for human data to provide a clear read on the ability of parenterally administered vaccines to protect the upper respiratory tract.

Reviewer #3:

Remarks to the Author:

Major claim: single dose vaccination either at the IN and oral route or muscle route provides complete protection against SARS-CoV-2 challenge. Data goes some way to supporting this, but only in the mouse model IN vaccine group is complete protection in the LRT and URT demonstrated. In light of this the claim of the title should be modified.

Article is well written and clear.

The findings in this manuscript are novel and of high interest to the community, and has the potential to influence the route of vaccination investigators in the field may consider.

There are several important issues which are not addressed in this body of work.

1. Description of the mouse model. Is there are reference outlining its development? Highlight this is mouse adapted SARS-CoV-2. BALB/c mice are not infected with WT virus.

2. Is this Ad5 SARS-CoV-2 vaccine currently under clinical evaluation? In Figure 1a four vaccine candidates are depicted. there needs to be an evidence based justification for choosing the construct which was advanced. It looks like there were 2 candidates which expressed well in Fig 1b. If available, please provide other references describing this vaccine.

3. It is of interest to know whether delivery route could bias a Th1:Th2 ratio? IgG1 vs. IgG2a ratio and /or IL-4/IL-13 T cell production should be investigated.

4. Expand upon rationale for dual IN and oral administration in the ferret model. Why not a single route of administration. If this was investigated please show data. Add further discussion on the feasibility of these routes of administration.

5. Only URT of the ferrets was examined. According to other publications the virus can infect the lungs of ferrets (Kim et al Cell Host Microbe 2020). Please discuss why LRT disease was not measured.

Minor points.

Needs a more comprehensive reference list.

Difficult to read type in figure 1.

Expand on the rationale for targeting both mucosal sites and muscle immunization. Discuss any potential qualitative differences in immune responses raised.

Reviewer #1 (Remarks to the Author):

The paper uses ferrets infected with SARS-COV2 and mice infected with a mouse adapted SARS-COV2 to show rAD5 Spike SAR2-cov2 vaccine-mediated protection against challenge. This is a nice story and results look compelling. I have a number of minor suggestions.

We highly appreciate the reviewer's comments and try to elucidate these comments.

1) The abstract should clarify that the mouse challenge used the mouse adapted virus, and that the ferret model (as I understand this) did not.

Answer: It has been clarified in the abstract (see line 20-22, page 1, in the revised manuscript)

2) l24 Fig 2 uses a single vaccination. Fig 3 uses IM or IN + oral. Not clear where the i.m. i.n. combination data referred to in the abstract is to be found? If this is speculation would seem inappropriate (given the data set) especially in the abstract.

Answer: There is no i.m. + i.n. combination data in the manuscript. We just investigated the single vaccination for the protection, including IM or mucosal vaccination (IN+oral), and found that mucosal vaccination had a better protective effect in this study. It has been clarified in the abstract (see line 23-25 in the revised manuscript).

3) The notion of i.n. vaccination is complicated by the risk of inducing asthma attacks. Although the experience for influenza is perhaps promising for this route of administration (Paediatr Respir Rev. 2014 Dec;15(4):340-7) this risk might be discussed.

Answer: Respiratory viruses are one of the most common causes of asthma exacerbations in both adults and children, however, the most commonly agents are rhinovirus and respiratory syncytial virus. Although intranasal vaccination with respiratory virus vector vaccine may increase the risk of asthma due to directly stimulating mucosal immunity, several studies have showed that live attenuated influenza vaccine given by nasal spray were safe and well-tolerated in children and adolescents with asthma. We recommend that people with a history of asthma not be vaccinated with this vaccine before it was proven to be safe for asthma patients. It has been discussed in the manuscript (see line 153-158 in the revised manuscript).

4) That preventing infection in the upper respiratory track is more difficult (l102) or easier in the lower respiratory track (l35) requires some clarity. More difficult in what sense? More antibodies needed? Virus replication not accessible to antibodies? T cells unable to engage MHC I? Higher vaccine doses needed to provide protection? Easier to access with vaccine? Higher replication capacity of virus? The concept is

useful perhaps but “difficult” and “easy” are very vague and difficult to understand.

Answer: In our work we found IM vaccination of Ad5-nCoV can prevent SARS-CoV-2 replication in the lungs of the mice, but not in the URT of the mice. In other’s work for the development of COVID-19 vaccines, we saw the similar phenomena. ChAdOx1 nCoV-19 vaccination by the intramuscular delivery can prevent SARS-CoV-2 replication completely in the lungs, but not in the URT of the rhesus macaques¹. Several kinds of DNA vaccines could not prevent SARS-CoV-2 replication completely in the URT tract of rhesus macaques². The similar results were also shown in the protective efficacy studies for the mRNA-1273 vaccine and the inactivated vaccines^{3,4}. So we think preventing virus replication in the URT is more difficult in the lower respiratory track. But in fact, we don’t know in which SARS-CoV-2 replicates more effectively, in the URT or the LRT.

In this work, we found mucosal vaccination can prevent virus replication in URT and LRT. We got some knowledge for the protection of the mucosal vaccination from the influenza virus. For the influenza nasal spray vaccines, IgA is the major mediator of immunity to influenza, and plasma IgG serves as a backup for IgA in the URT, whereas in the LRT, IgG is the dominant antibody involved in protection⁵. We speculate IgA is also the major mediator of immunity to SARS-CoV-2 and T cell response could be also very important for clearance of the virus, which will be elucidated in the future work.

In current animal models we think higher dose of vaccine cannot increase the protection because the vaccination dose is high enough for ferrets (one human dose) and mice (1/10 human dose). Both intramuscular delivery and mucosal delivery of Ad5-nCoV can stimulate the immune system efficiently in these animal models.

1. Doremalen NV, *et al.* 2020. ChAdOx1 nCoV-19 vaccination prevents SARS-CoV-2 pneumonia in rhesus macaques. *bioRxiv* preprint doi: <https://doi.org/10.1101/2020.05.13.093195>.
2. Yu J, *et al.* 2020. DNA vaccine protection against SARS-CoV-2 in rhesus macaques. *Science* 10.1126/science.abc6284 (2020).
3. Corbett KS, *et al.* 2020. SARS-CoV-2 mRNA Vaccine Development Enabled by Prototype Pathogen Preparedness. *bioRxiv* preprint doi: <https://doi.org/10.1101/2020.06.11.145920>.
4. Gao Q, *et al.* 2020. Development of an inactivated vaccine candidate for SARS-CoV-2. *Science*. 10.1126/science.abc1932 (2020).
5. Broadbent AJ, *et al.* 2015. Chapter 59 - Respiratory Virus Vaccines. *Mucosal Immunology (Fourth Edition)*.1, 1129-1170.

5) What is the rationale for in + oral? Oral vaccination needs some methodological explanation as quite an unusual route of entry? Any evidence that oral actually takes (works) in ferrets? What happens with i.n alone?

Answer: Mucosal immune system is big value against the infection of SARS-CoV-2. However, it is well known that the adaptive mucosal immunity usually is hard to

induce and to keep lasting. Many studies had suggested there is a huge and powerful lymphoid tissue system in the whole mucosal of digestive tract. Oral administration of Adenovirus 5-vectored and other vaccines successfully protects ferrets and other animals against multiple viruses, and oral vaccination of modified live rabies vaccine induce long lasting protective immunity¹⁻⁵. We speculated that combined i.n. and oral vaccination with Ad5-nCoV may elicit stronger and lasting immunity in ferrets.

1. Scallan, C. D., Lindbloom, J. D. & Tucker, S. N. Oral modeling of an adenovirus-based quadrivalent influenza vaccine in ferrets and mice. *Infectious diseases and therapy* **5**, 165-183, doi:10.1007/s40121-016-0108-z (2016).
2. Brown, L. J. *et al.* Oral vaccination and protection of striped skunks (*Mephitis mephitis*) against rabies using ONRAB(R). *Vaccine* **32**, 3675-3679, doi:10.1016/j.vaccine.2014.04.029 (2014).
3. Fry, T. L., Vandalen, K. K., Duncan, C. & Vercauteren, K. The safety of ONRAB(R) in select non-target wildlife. *Vaccine* **31**, 3839-3842, doi:10.1016/j.vaccine.2013.06.069 (2013).
4. Sobey, K. G. *et al.* ONRAB(R) oral rabies vaccine is shed from, but does not persist in, captive mammals. *Vaccine* **37**, 4310-4317, doi:10.1016/j.vaccine.2019.06.046 (2019).
5. Shuai, L. *et al.* Genetically modified rabies virus-vectored Ebola virus disease vaccines are safe and induce efficacious immune responses in mice and dogs. *Antiviral research* **146**, 36-44, doi:10.1016/j.antiviral.2017.08.011 (2017).

6) 140 perhaps rephrase “has shown a favourable safety profile”

Answer: It has been rephrased according to your suggestion (see line 38-39 in the revised manuscript).

7) 148 Very unclear what prior knowledge is referring to here, as no references provided. The rationale might be explained so that the reader can understand how this guiding worked.

Answer: We developed several vaccines based on Ad5 vector vaccines, such as Ebola virus vaccine¹, Marburg virus vaccine² and Zika virus vaccine³. Increasing the antigen expression is most important for the design of each vaccine. The codon optimization including the optimization of GC content of the polynucleotides chain, and the signal peptide selection are critical for the antigen expression. The references have been added (see line 46 in the revised manuscript).

1. US patent, US10172932, Ebola virus disease vaccine taking human replication deficient adenovirus as vector.
2. China patent, ZL201810428286.1, Marburg virus disease vaccine with human replication deficient adenovirus as vector.
3. Guo, Q. *et al.* Immunization with a novel human type 5 adenovirus-vectored vaccine expressing the premembrane and envelope proteins of Zika virus provides consistent and sterilizing protection in multiple immunocompetent and

immunocompromised animal models. *The Journal of infectious diseases* 218, 365-377, doi:10.1093/infdis/jiy187 (2018)

8) *Many have moved to chimp adenovirus because of pre-existing immunity to Ad5 – this needs at the very least to be discussed.*

Answer: Pre-existing immunity to Ad5 is really a limitation of the Ad5 vector vaccine due to the high seroprevalence of this vector. High pre-existing immunity weakened the humoral and cellular immune response on the HIV, Ebola and COVID-19 vaccine. Other evidence suggests that non injectable vaccination, such as nasal delivery or sublingual delivery, of Ad5-based vaccine can bypass pre-existing immunity to the vaccine carrier in animal models, and oral adenoviral delivery of influenza vaccine induced both systemic and mucosal immune responses in human. These studies suggest that mucosal vaccination of Ad5-vectored vaccines can induce good immune response to avoid pre-existing immunity to the vaccine carrier. It has been discussed in the manuscript (see line 149-153 in the revised manuscript)

9) *The MWU test assumes equal variance – a condition not met in several of the Figs eg Fig. 3 d and e where the red circles are all below detection (i.e. zero variance) and are compared with data sets that clearly have variance.*

Answer: Two-tailed nonparametric Mann-Whitney's rank tests were conducted to compare differences between 2 experimental groups, and Kruskal-Wallis ANOVA with Dunn's multiple comparisons tests were applied to compare >2 experimental groups. Antibody titre data were log transformed before analysis. The data were re-analyzed and some modifications are made to the statistical results [see Figure 1i, 1j (line 247), Figure 2d, 2e (line 269), Extended Data Figure 6 (line 534), and lines 98, 122 and 472-496 in the revised manuscript).

10) *Would be helpful if the rAd5 doses used in humans were discussed to address whether the Ad5 doses used in the animal models might be viewed as appropriate?*

Answer: We explored the vaccination dose in the mouse model but not in ferret model due to the limited number of ferrets. The dose there in ferret model is equal to one human dose in clinical trial, which is high enough for ferrets. The dose dependent immune response was found in mouse model, however all the intranasal vaccination mouse achieved sterile protection against SARS-CoV-2. The low dose there in mouse model is equal to 1/1000 human dose (5×10^{10} vp) in clinical trial. The dose in the phase 1 human clinical trial was set according to the dose in our Ebola vaccine clinical trial, including three dose groups, 5×10^{10} vp, 10×10^{10} vp and 15×10^{10} vp.

11) *1135 These were vaccine-induced tissue resident T cells – not just tissue resident T cells.*

Answer: It has been corrected according to your suggestion (see line 147 in the revised manuscript).

12) Did the lack of cellular immune responses 1143/4 correlate with increased post change viral titers? Does this provide insights into the relative protective value of anti-viral T vs ab responses?

Answer: Both neutralizing antibody and T cell responses were clearly important in eliminating the virus and controlling disease development in the COVID-19 patients who were naturally infected by SARS-CoV-2. Antibodies are effective against SARS-CoV-2 because several monoclonal antibodies against SARS-CoV-2 were illuminated their protection in the animal models^{1, 2}. T cell response in protection against SARS-CoV-2 is still not to be elucidated until now. But for the vaccine-induced immune responses, whether neutralizing antibody alone is capable of preventing infection remains undetermined. Specific T cell responses are essential for directly attacking and killing virus-infected cells. In addition, the CD4 T cells responses are critical for the cytotoxic T-cell response and the maturing of neutralizing antibodies. Thus, the evaluation of the cellular mediated responses besides the neutralizing antibodies, is important for a candidate vaccine.

1. Shi R, *et al.* A human neutralizing antibody targets the receptor binding site of SARS-CoV-2. *Nature*. Doi:10.1038/s41586-020-2381-y(2020).
2. Cao Y, *et al.* Potent neutralizing antibodies against SARS-CoV-2 identified by high-throughput single-cell sequencing of convalescent patients' B cells. *Cell*. 182,1-12. Doi: 10.1016/j.cell.2020.05.025.(2020)

Reviewer #2 (Remarks to the Author):

Comments for the Author

The Manuscript by Wu et al., describes the protective efficacy of Ad5-nCov vaccine against SARS-CoV-2 in mouse and ferret models. The current study demonstrated that upper respiratory tract (URT) administration of Ad5-nCov could achieve sterile protection against SARS-CoV-2 in both animal models. The manuscript also addresses a highly relevant point regarding the influence of vaccination route (i.e. IM vs IN route). The current study demonstrated that the vaccine administration to the URT may be more effective in inducing immune responses at the site of SARS-CoV-2 infection. Overall, the work is well done and well presented. These data will provide important information to those currently developing SARS-CoV-2 vaccines.

We highly appreciate the reviewer's comments and try to elucidate these comments.

Specific comments;

1: Authors should clarify the type of deletion that leads to the defect in replication in the results of the manuscript. Presumable this is E1/E3 deleted Admax?

Answer: It has been clarified in the results of the manuscript (see line 49-50 in the revised manuscript).

2: The titers of neutralizing antibodies shown here for mice and ferrets are well above those published in their Lancet paper. This should be discussed as well as the potential impact of relative dose in humans compared to small animals.

Answer: The titers of neutralizing antibodies in the high and middle-dose groups of mice and ferrets are higher than those of the subjects in the phase I clinical trial of Ad5-nCoV. The dose for ferrets and the high and middle doses for mice were much higher than human dose if calculated by body weight or body area ratio. The GMT of the neutralizing antibody for the low dose group is 23 (IM) and 19 (IN), which is comparable to those of the subject in the human clinical trial on day 28 post vaccination. It has been added in the discussion of the manuscript (see line 129-133 in the revised manuscript).

3. It is probably worth mention in the discussion that a mucosal vaccine that protects against URT replication may have important advantages over parenterally administered vaccines with regard to interrupting person to person transmission.

Answer: This is a very good suggestion, we have added the comment in discussion (see line 139-145 in the revised manuscript).

4. Line 131-“Whether a lower challenge dose would result.....”-this may be true but the opposite result is equally likely. Although the natural dose one might encounter is lower, the human is more permissive to SARS-CoV-2. It is likely worth noting both possibilities and that we need to wait for human data to provide a clear read on the ability of parenterally administered vaccines to protect the upper respiratory tract.

Answer: Very good suggestion. Although the natural dose one might encounter is much lower than that of animals in a challenge study, the human is more permissive to SARS-CoV-2. Whether the IM vaccination or mucosal vaccination would protect the URT and LRT in humans under natural exposure remains to be determined in phase III trials. We have added some of the comments in the discussion (see line 135-139 in the revised manuscript).

Reviewer #3 (Remarks to the Author):

We highly appreciate the reviewer’s comments and try to elucidate these comments.

Major claim: single dose vaccination either at the IN and oral route or muscle route provides complete protection against SARS-CoV-2 challenge. Data goes some way to supporting this, but only in the mouse model IN vaccine group is complete protection in the LRT and URT demonstrated. In light of this the claim of the title should be modified.

Answer: The title has been changed to “A single dose of an adenovirus-vectored vaccine provides complete protection against SARS-CoV-2 challenge”.

Article is well written and clear.

The findings in this manuscript are novel and of high interest to the community, and has the potential to influence the route of vaccination investigators in the field may consider.

There are several important issues which are not addressed in this body of work.

1. Description of the mouse model. Is there are reference outlining its development? Highlight this is mouse adapted SARS-CoV-2. BALB/c mice are not infected with WT virus.

Answer: SARS-CoV-2 virus do not efficiently infect BALB/c mice. As shown in Fig.2, however, the mouse-adapted SARS-CoV-2 virus that was generated by passaging in BALB/c mice and was used in this study, replicated well in nasal turbinates and lungs of mice. We briefly described the mouse-adapted virus in Methods (lines 290-294) and Main text (line 87). Now we have highlighted the “mouse-adapted” in Fig. 2 legend (line 264).

2. Is this Ad5 SARS-CoV-2 vaccine currently under clinical evaluation? In Figure 1a four vaccine candidates are depicted. There needs to be an evidence based justification for choosing the construct which was advanced. It looks like there were 2 candidates which expressed well in Fig 1b. If available, please provide other references describing this vaccine.

Answer: This vaccine is currently under a phase I and a phase II clinical evaluation. There were 2 candidates expressed well in Fig 1b, and the expression level of “SP_{IPA} add S_{optimized}” (lane 4) was higher than that of “SP_{original} add S_{optimized}” (lane 3) in transfected cells by western blot in the repeated tests. We further investigated the specific anti-S IgG titers for Ad5-SPori-Sopt and Ad5-nCoV in mice and the specific IgG GMT of Ad5-SPori-Sopt is significantly lower than that of Ad5-nCoV at day 28 post immunization(116054 vs 163003, P=0.0052) (data not shown in the manuscript). So the Ad5-nCoV was chosen for the further development.

3. It is of interest to know whether delivery route could bias a Th1:Th2 ratio? IgG1 vs. IgG2a ratio and /or IL-4/IL-13 T cell production should be investigated.

Answer: The IgG1 and IgG2a titres in serum of the IM and IN vaccination mouse at week 8 post immunization had been detected. Both the IM and IN route can induce the robust SARS-CoV-2 S specific IgG1 and IgG2a responses and the IM route can achieve a significantly higher IgG2a / IgG1 ratio. It demonstrated that the IM route induced a higher Th1 response than the IN route. The results have been added to the manuscript (see line 65-67 and line491-499 in the revised manuscript).

The Th2 cytokines, IL-4/IL-13 T cell production were not detected in the present

study. But we investigated IL-4 in the Ad5-nCoV-vaccinated cynomolgus monkeys by intramuscular delivery. No IL-4 was produced, but IFN- γ , TNF- α and IL-2 were significantly produced in the Ad5-nCoV-vaccinated cynomolgus monkeys, which showed the higher Th1 response. IL-4 secreted by T cell by the mucosal vaccination will be investigated in the following work.

4. *Expand upon rationale for dual IN and oral administration in the ferret model. Why not a single route of administration. If this was investigated please show data. Add further discussion on the feasibility of these routes of administration.*

Answer: Mucosal immune system is big value against the infection of SARS-CoV-2. However, it is well known that the adaptive mucosal immunity usually is hard to induce and to keep lasting. Many studies had suggested there is a huge and powerful lymphoid tissue system in the whole mucosal of digestive tract. Oral administration of Adenovirus 5-vectored and other vaccines successfully protects ferrets and other animals against multiple viruses, and oral vaccination of modified live rabies vaccine induce long lasting protective immunity¹⁻⁵. We speculated that combined i.n. and oral vaccination with Ad5-nCoV may elicit stronger and lasting immunity in ferrets. In this study, we demonstrated that mucosal vaccination of Ad5-nCoV via combination of IN and oral routes can protect ferrets against SARS-CoV-2 challenge. The protective efficacy of vaccination via IN or oral route alone will be investigated in the future study.

1. Scallan, C. D., Lindbloom, J. D. & Tucker, S. N. Oral modeling of an adenovirus-based quadrivalent influenza vaccine in ferrets and mice. *Infectious diseases and therapy* **5**, 165-183, doi:10.1007/s40121-016-0108-z (2016).
2. Brown, L. J. *et al.* Oral vaccination and protection of striped skunks (*Mephitis mephitis*) against rabies using ONRAB(R). *Vaccine* **32**, 3675-3679, doi:10.1016/j.vaccine.2014.04.029 (2014).
3. Fry, T. L., Vandalen, K. K., Duncan, C. & Vercauteren, K. The safety of ONRAB(R) in select non-target wildlife. *Vaccine* **31**, 3839-3842, doi:10.1016/j.vaccine.2013.06.069 (2013).
4. Sobey, K. G. *et al.* ONRAB(R) oral rabies vaccine is shed from, but does not persist in, captive mammals. *Vaccine* **37**, 4310-4317, doi:10.1016/j.vaccine.2019.06.046 (2019).
5. Shuai, L. *et al.* Genetically modified rabies virus-vectored Ebola virus disease vaccines are safe and induce efficacious immune responses in mice and dogs. *Antiviral research* **146**, 36-44, doi:10.1016/j.antiviral.2017.08.011 (2017).

5. *Only URT of the ferrets was examined. According to other publications the virus can infect the lungs of ferrets (Kim et al Cell Host Microbe 2020). Please discuss why LRT disease was not measured.*

Answer: SARS-CoV-2 virus replicates efficiently in URT of the ferrets. However, the vRNA copies and infectious virus titers in the lungs of ferrets were very low (Kim et al Cell Host Microbe 2020) or undetectable (Shi et al Science 2020), although

pathological changes were observed in lungs. Thus, we investigated both the URT and LTR diseases in mice, but only examinations in URT in ferrets. Now we have changed the description on lines 104 and 105.

Minor points.

Needs a more comprehensive reference list.

Answer: We have added several comments in discussion and 11 references have been added at the same time (see references in the revised manuscript).

Difficult to read type in figure 1.

Answer: The type in figure 1 has been revised (see figure 1 (line 246) in the revised manuscript).

Expand on the rationale for targeting both mucosal sites and muscle immunization.

Answer: Originally we thought that combination vaccination had some advantages, including rapid and strong systemic immunity induced by IM vaccination and mucosal immunity at the entry port induced by mucosal vaccination. But considering the complexity in the practical applications, we believe that the mucosal vaccination is more advantageous than the IM vaccination. So the statement of “combination vaccination” is removed in the revised manuscript.

Discuss any potential qualitative differences in immune responses raised.

Answer: The discussion was added into the manuscript (see 139-145 in the revised manuscript).